# Inflammatory Undifferentiated Pleomorphic Sarcoma Mimicking Bacteremia in an Elderly Patient: A Case Report

**DOI:** 10.3390/medicina57020175

**Published:** 2021-02-18

**Authors:** Kazuhiko Hashimoto, Shunji Nishimura, Tomohiko Ito, Naohiro Oka, Masao Akagi

**Affiliations:** Department of Orthopedic Surgery, Kindai University Hospital, Osaka-Sayama City, Osaka 589-8511, Japan; shunnisi@med.kindai.ac.jp (S.N.); tomo0251118zooo@gmail.com (T.I.); n-oka@med.kindai.ac.jp (N.O.); makagi@med.kindai.ac.jp (M.A.)

**Keywords:** Bacteremia, C-reactive protein, inflammation, interleukin 6, undifferentiated pleomorphic sarcoma

## Abstract

Undifferentiated pleomorphic sarcoma (UPS) is major type of soft tissue sarcomas. UPS presenting with inflammation is rare, and its pathophysiology remains unclear. Herein, we report a rare case of UPS with prolonged fever. A 91-year-old female complaining of high fever was referred to our hospital because of a high C-reactive protein (CRP) level of 12.51 mg/dL. She had been experiencing intermittent fevers for approximately 10 years. The fever of unknown origin worsened with time and went into remission with repeated antimicrobial therapy. She also had a mass on her central lower back over the sacral region for 6 years, which showed a gradual increase in size. The blood tests showed that the leukocyte count and neutrophils were 6.51 × 10^3^ /µL and 70.3%, respectively. She had a 10 × 10 cm mass on her buttock that showed 2-[fluorine-18] fluoro-2-deoxy-d-glucose (FDG) accumulation on FDG-positron emission tomography-computed tomography examination (standardized uptake value-max value: 5.4). A blood culture examination was performed to rule out bacteremia, however, no bacteria were identified. We then performed a needle biopsy and confirmed the diagnosis of UPS; subsequently, the patient underwent a wide-margin resection. A few days after the surgery, her CRP, leukocyte, and neutrophil levels decreased to 0.305 mg/dL, 2.83 × 10^3^/uL, and 50.1%, respectively. This case demonstrated that UPS with inflammation should be treated surgically as soon as possible after ruling out other sources of infection to achieve a favorable prognosis.

## 1. Introduction

There are numerous histological types of soft tissue sarcomas (STSs), and more than 60 types have been defined [1]. The malignant fibrous histiocytoma (MFH) that is an ordinary STS, was first described in 1964 [2]. MFH is a mesenchymal neoplasm that is anatomically ubiquitous and occurs in all ages [2]. In 2002, the World Health Organization (WHO) classification of STS defined MFH as undifferentiated pleomorphic sarcoma (UPS) [1]. Recently, MFH and UPS associated with inflammation have been reported; however, the details of their pathogenesis remain unknown. [3,4]. Herein, we report a case of UPS with severe inflammation in a very elderly patient.

## 2. Case Presentation

The patient was a 91-year-old woman. She had been experiencing fever of unknown origin intermittently for approximately 10 years. She also had a tumor in the central lower back over the sacral region for 6 years. Recently, during hospitalization in another center, she was being treated with antibiotics and blood infusion for high fever and anemia. However, the fever did not improve, and she was referred to our hospital.

The blood examination revealed C-reactive protein (CRP) level of 12.51 mg/dL, white blood cell count of 6.51 × 10^3^/uL, and neutrophils at 70.3% (Table 1).

We performed a blood culture test that revealed no evidence of bacteremia. Radiography of the chest showed no infiltrative shadow suggestive of pneumonia. Urine culture revealed no evidence of bactereia. The 10 x 10 cm tumor mass on her buttock was elastic, hard, and immobile. Magnetic resonance imaging (MRI) showed a low-intensity mass adjacent to the sacrum bone on T1-weighted and low- and high- intensity areas on T2-weighted images (Figure 1a,b, respectively). 2-[fluorine-18] fluoro-2-deoxy-d-glucose (FDG)-positron emission tomography (PET)-computed tomography (CT) examination also revealed the tumor mass on her buttock, for which the standardized uptake value (SUV)-max value was 5.4 (Figure 1c). No suspected metastatic lesions, such as in the lungs, were observed. We then performed a needle biopsy of the buttock tumor. The histology showed an increased amount of spindle cells stained well with atypical nuclei (Figure 1d). Heterozygous cells had darkly stained irregular or oblong heterozygous nuclei and stellate or spindle-shaped cytoplasm (Figure 1d). Immunohistological staining for CDK4, MDM2, S-100, SMA, desmin, HHF-35, myogenin, Myf4, and MyoD1 were also negative (data not shown). Since the differentiation tendency of the tumor proved difficult to determine, the patient was finally diagnosed with UPS based on these histological findings.

Subsequently, the patient underwent wide-margin resection. The surgical margin of the resected specimen was microscopically positive [5]. Lymphocytic infiltration was observed within the tumor (Figure 1e) and at the tumor margins (Figure 1f). Further, we observed an abundance of blood vessels in the tumor (Figure 2a). Some megakaryocytes were also observed among the tumor cells (Figure 2b).

Immunostaining was partially positive for smooth muscle antigen and slightly positive for myogenin (Figure 2c,d, respectively). The Ki-67 labeling index was approximately 50% (Figure 2e). Immunostaining was negative for caldesmon, calponin, desmin, S-100, CD34, CD31, MDM2, and CDK4 (data not shown). The diagnosis of leiomyosarcoma was excluded because SMA was only partially positive, and desmin and HHF-35 were negative [6,7]. Furthermore, the diagnosis of rhabdomyosarcoma was excluded since myogenin was just slightly positive, and MyoD1, desmin, and HHF-35 were negative [6,8]. The diagnose of synovial sarcoma and malignant peripheral sheath tumor were also excluded because S-100 was negative [9]. Finally, we confirmed the diagnosis of UPS based on the histological findings of HE staining and these excluded diagnoses.

However, the immunostaining results were negative for CDK4, MDM2, CD31, CD34, S-100, desmin, AE1/AE3, Myf4, and calponin (data not shown). The inflammation markers of the blood improved a few days after the surgery (Figure 3); the patient’s postoperative CRP level, white blood cell count, and neutrophil count were 0.305 mg/dL, 2.83 × 10^3^/µL, and 57.9%, respectively. No recurrence was observed at one year after surgery.

## 3. Discussion

Inflammatory malignant fibrous histiocytoma (IMFH) is a rare neoplasm and was first reported in 1976 [10]. Approximately 22–65% of MFH cases are CRP positive and show inflammatory infiltrates without infection, as previously described [3,4,11]. In the previously reported case series of 11 IMFH cases, positive CRP was not reported to be as high (<1.0 mg/dL) as in our case [3]. Recently, according to the WHO classification guidelines for STS, the MFH classification was eliminated in 2002 and was replaced with UPS [1]. Hayashida et al. reported that highly inflammatory UPS (IUPS) (CRP = 11.07 mg/dL, leukocyte = 69,100/µL) causes leukemoid reaction [4], and in the past, only one case of highly IMFH or IUPS was reported in the buttocks [12].

Over the last few years, major advances have been made in the characterization of the tumor microenvironment of soft tissue sarcoma, with the description of “hot tumors” massively infiltrated by immune cells and “cold tumors” with no significant immune infiltration [13]. Moreover, Petitprez, et al. established an immune-based classification on the basis of the composition of the tumor microenvironment and identified five distinct phenotypes: immune-low (A and B), immune-high (D and E), and highly vascularized (C) [13]. The report also indicated that the class-E group demonstrated improved survival and a high response rate to PD1 blockade with pembrolizumab in a phase 2 clinical trial [13]. In our case, blood vessels were abundant in the tumor, and lymphocytic infiltration was noted within the tumor, and at the tumor margins.

These findings suggested that the tumor was a “hot tumor,” and there was a relationship between the tumor microenvironment and systemic inflammation. The current case could be further classified as immune-high (D and E), and/or highly vascularized (C) groups. Rapid growth and early metastasis have been observed in IUPS, and the involvement of G-CSF, IL-6, IL-7, IL-8, SCF, TGB, and G-CSF have also been suggested [4,14,15]. IL-6 is an important cytokine that plays various roles in many cells, such as proliferation of T lymphocytes, induction of B lymphocyte maturation, promotion of the growth of erythroid cells, myeloid cells, and megakaryocytes, and the induction of acute phase protein production by hepatocytes [16,17,18]. In this case, intratumoral infiltration of megakaryocytes and lymphocytes was observed. Thus, it was suggested that inflammatory cytokines might be acting on the tumor microenvironment.

Diagnostic delays for STS have been reported previously and remain a huge issue that require better patient and doctor awareness [19]. Cancer diagnosis in the elderly is often delayed due to patient dementia [20]. In the current case, although the patient had no dementia, the diagnosis was delayed because the patient herself did not visit the clinic. However, the fever and the inflammatory reaction prompted her to visit the doctors, who were then able to find the sarcoma.

High uptake of FDG indicates higher tumor cellularity, biological behaviors of the tumor cells, and the composition and proportion of inflammatory cells [21]. Moreover, FDG uptake indicates marked leukocytosis in IUPS [4]. In the current case, although the patient showed a high level of inflammation, we observed a relatively low SUV-max; the mean SUV-max value, in general, is reported to be 10.78 ± 6.72, as described previously [22]. The heterogeneous FDG uptake suggested large necrotic areas.

The basic treatment for STS is wide-margin resection [1]. STS with inflammation or highly aggressive histology should be treated with neoadjuvant chemotherapy or radiation therapy [23,24]. However, neoadjuvant therapy is often difficult for the elderly because they have decreased immunity [25]. Therefore, an adequate wide margin should be acquired by surgery to obtain a favorable prognosis in such STS patients [26]. In the current case, although we planned to acquire wide-margin resection, the margin of the resected specimen was inadequate (R1) [5]. Despite inadequate surgical margins, additional radiation therapy and surgical resection were not performed because the patient and his family did not wish to undergo them. A previous study showed that pretreatment CRP or IL-6 levels are independent prognostic factors for STS [3,27]; however, whether old age is itself a prognostic factor is still controversial [28,29]. The STS recurrence generally occurs within the first two years after surgery [30]. Thus, careful follow-up is necessary in the future.

The present study had some limitations. First, the excluded diagnoses were due to the absence of an in-situ hybridization specific test for each tumor [31,32,33].

However, we were able to make a definitive diagnosis with HE staining and immunostaining.

In addition, we did not measure the blood levels of systemic inflammatory cyto-kines such as IL-6.

However, we were able to suggest the possibility of an association between tumor microenvironment and systemic inflammation from the histopathology. Moreover, we did not investigate several immune molecules, such as CD3, CD8, CD20, CD34, PD-1, and PD-L1, to confirm the immune-based classification [13]. If this tumor could have been classified as category E, PD-1 inhibitor therapy might have been indicated. This could be an interesting subject for future research.

## 4. Conclusion

In conclusion, we encountered a rare case of IUPS in a very elderly patient with high CRP. IUPS can be diagnostically confused with other infections, such as bacteremia, and even if there is no indication of bacteremia, blood cultures should be collected immediately to rule out infection. In addition, after ruling out such infections, prompt wide-margin resection should be performed to obtain a good patient prognosis.

## Figures and Tables

**Figure 1 medicina-57-00175-f001:**
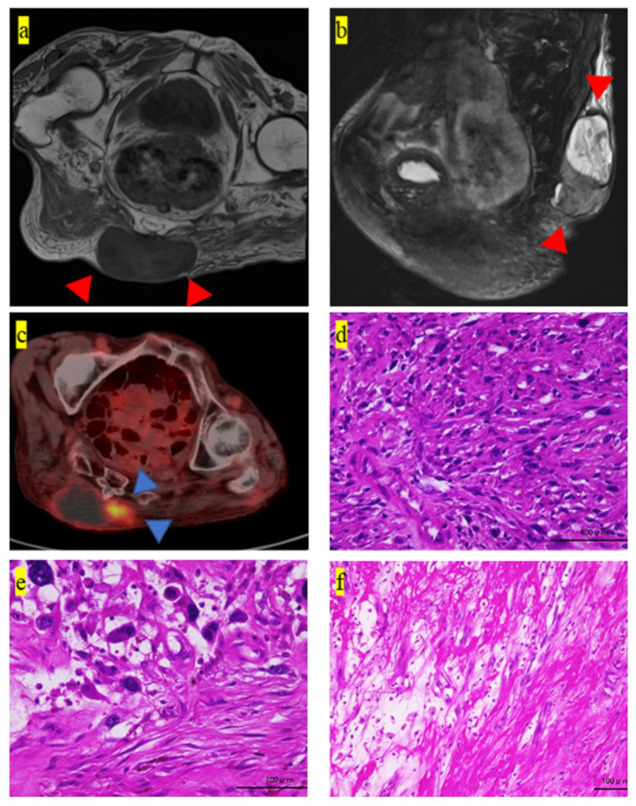
Pathological and imaging findings of the excised tumor. (**a**) T1-weighted MRI coronal image at the first patient visit. Homogeneous low-intensity tumor mass is observed on the buttock (red arrowheads). (**b**) T2-weighted MRI sagittal image at the first patient visit. Heterogeneous low- and high-intensity areas in the tumor mass are observed (red arrowheads). (**c**) 18F-fluorodeoxyglucose-positron emission tomography-CT imaging shows heterogeneous accumulation in the tumor (blue arrowheads). (**d**–**f**) The pathological findings of the excised specimen. The histology shows (**d**) increased spindle cells stained well with atypical nuclei, (**e**) lymphocytes infiltration in the tumor, and (**f**) lymphocytes infiltration of tumor margins. Scale bar = 100 um. MRI, Magnetic resonance imaging; CT, computed tomography.

**Figure 2 medicina-57-00175-f002:**
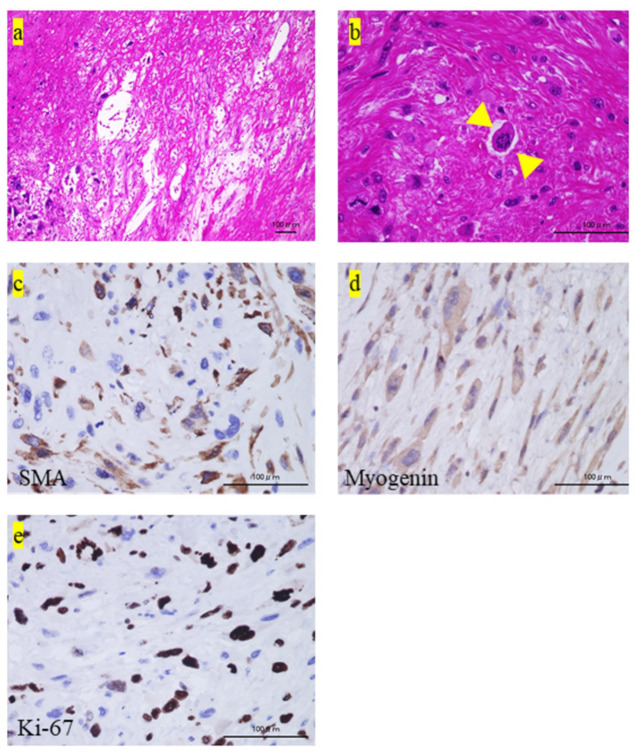
Pathological and iImmunohistochemical findings of the tumor. (**a**,**b**) The pathological findings of the excised specimen. The histology shows (**a**) increased vascular penetration into the tumor, (**b**) megakaryocytes (yellow arrowheads). (**c**–**e**) The immunohistochemical findings show positive staining for (**c**) smooth muscle antigen (SMA), (**d**) myogenin, (**e**), and Ki-67. IL, Interleukin.

**Figure 3 medicina-57-00175-f003:**
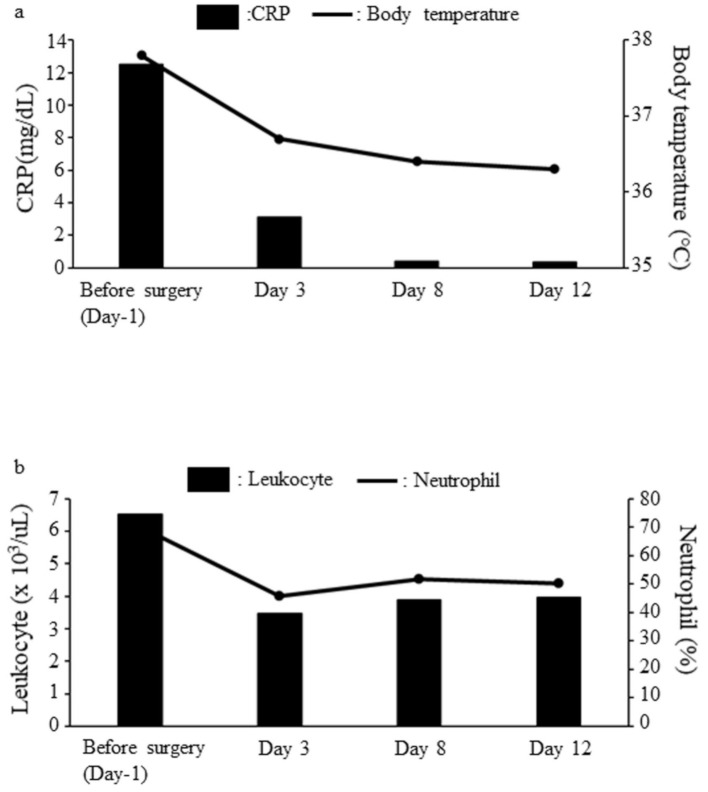
Pre- and post-operative comparison of inflammatory markers. (**a**) The pre- and post-operative progress chart of the CRP and body temperature. The black box shows CRP (mg/dL) and the black line shows body temperature (°C). The CRP level and body temperature decrease and reach a normal value 3 days after surgery. (**b**) The pre- and post-operative progress chart of leukocytes (x 10^3^/µL) and neutrophils (%). The black box shows leukocytes (x 10^3^/µL), and the black line shows neutrophils (%). The leukocytes and neutrophils decreased and become close to normal values 3 days after surgery. CRP, C-reactive protein.

**Table 1 medicina-57-00175-t001:** The results of blood examination before the surgical treatment.

Item	Value	Lower Limit	Upper Limit
**CRP (mg/dL)**	12.51	0	0.14
**Cr (mg/dL)**	1.28	0.46	0.79
**Alb (g/dL)**	2.1	4.1	5.1
**AST (U/L)**	26	13	30
**ALT (U/L)**	23	7	23
**Leukocytes (x 10^3^/µL)**	6.51	3.3	8.6
**Hb (g/dL)**	7.9	11.6	14.8
**PLT** **(x 10^4^/uL)**	38.0	15.8	34.8
**Neutrophils (%)**	70.3	38	77

CRP, C-reactive protein; Cr, creatinine; Alb, Albumin; AST, Aspartate aminotransferase; ALT, Alanine aminotransferase; Hb, Hemoglobin; PLT, Platelet.

## Data Availability

The datasets used and/or analyzed during the current study are available from the corresponding author on reasonable request.

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
