# Peer review of "Inflammatory Undifferentiated Pleomorphic Sarcoma Mimicking Bacteremia in an Elderly Patient: A Case Report"

_medicina, 2021, doi:10.3390/medicina57020175_

Round 1

Reviewer 1 Report

In this paper, the authors report a case of UPS associated with systemic inflammation, as reflected by fever and high levels of serum CRP and white blood cells. Inflammation disappeared after surgical resection of the primary tumor.

The report is interesting but faces several limits that should be answered by the authors:

Major comments:

-Over the last few years, major advances have been made in the characterization of the tumor microenvironment of soft tissue sarcoma, with the description of "hot tumors" massively infiltrated by immune cells and "cold tumors" with no significant immune infiltration (Petitprez et al Nature 2020). Are there any clues suggesting that systemic inflammation might be associated to a specific pattern of tumor microenvironment, and how looked the TME like in this particular case?

- the pathological description of the tumor lacks precision and détails; can any other other diagnosis such as leiomyosarcoma and rhabdomyosarcoma by excluded? Were  any molecular analysis performed (ISH?)

- PET-CT is not a standard procedure for STS. Moreover, Figure 1 shows a heterogeneous FDG uptake suggesting large necrotic areas rather than low metabolic activity. TEP-FDG is not a gold standard procedure to rule out bacteraemia (as specified in the conclusion)

- The reason why  IL6 immunostaining is unclear. Systemic inflammation should be associated to high systemic IL6 serum levels. Was this test performed in this case? Any clues for a link between high systemic IL6 serum levels and tumor expression?

Minor comments

The manuscript needs English language reediting

Units for CRP and blood cells should homogenized in standard format

Author Response

Thank you very much for reviewing this paper.

I appreciate your comments, which have helped me in making corrections to the document.

Deletions are shown in red and horizontal lines, and additions are shown in blue.

We believe that we have made a better paper.

I hope you will review it again.

Reviewer#1

Comments and Suggestions for Authors

In this paper, the authors report a case of UPS associated with systemic inflammation, as reflected by fever and high levels of serum CRP and white blood cells. Inflammation disappeared after surgical resection of the primary tumor.

The report is interesting but faces several limits that should be answered by the authors:

Major comments:

-Over the last few years, major advances have been made in the characterization of the tumor microenvironment of soft tissue sarcoma, with the description of "hot tumors" massively infiltrated by immune cells and "cold tumors" with no significant immune infiltration (Petitprez et al Nature 2020). Are there any clues suggesting that systemic inflammation might be associated to a specific pattern of tumor microenvironment, and how looked the TME like in this particular case?

Author’s response: Thank you for pointing out this important question. In the current case, we could observe infiltration of lymphocytes in the tumor and in the border between the tumor and surrounding tissues. Many vascular penetrations were also observed. These findings suggest that the current tumor is “hot tumor” and systemic inflammation might be associated with the immune cells infiltrating tumor microenvironment.

Author’s action:

We added these sentences in the discussion part as shown below;

Over the last few years, major advances have been made in the characterization of the tumor microenvironment of soft tissue sarcoma, with the description of "hot tumors" massively infiltrated by immune cells and "cold tumors" with no significant immune infiltration [14]. In this case, blood vessels were abundant in the tumor, lymphocytic infiltration within the tumor, and at the tumor margins.

These findings suggested that the tumor was a “hot tumor,” and there was a relationship between the tumor microenvironment and systemic inflammation (Line 125-131).

- the pathological description of the tumor lacks precision and détails; can any other other diagnosis such as leiomyosarcoma and rhabdomyosarcoma by excluded? Were  any molecular analysis performed (ISH?)

Author’s response:

Thank you for pointing out this problem.

We agree with your observation that we have not been able to describe the process of definitive diagnosis.

We carefully examined the immunostaining results again and confirmed that SMA was only partially positive and myogenin was only slightly positive, not specifically positive.

We ruled out leiomyosarcoma because the immunostaining was negative for desmin and HHF-35, and SMA was only partially positive.

We also excluded rhabdomyosarcoma because desmin and HHF-35 were negative, myogenin was slightly positive, and myoD1 was negative.

We did not perform any molecular analysis, including ISH. We think this point is a limitation of the current study; therefore, we mentioned this in the limitation part.

Author’s action:

We have revised these sentences in the case presentation part as shown below:

Immunostaining was partially positive for smooth muscle antigen and slightly positive for myogenin (Figure 2c and 2d). The Ki-67 labeling index was approximately 50% (Figure 2e). Immunostaining was negative for caldesmon, calponin, desmin, S-100, CD34, CD31, MDM2, and CDK4 (data not shown). The diagnosis of leiomyosarcoma was excluded because SMA was only partially positive, and desmin and HHF-35 were negative [7, 8]. Furthermore, the diagnosis of rhabdomyosarcoma was excluded since myogenin was just slightly positive, and MyoD1, desmin, and HHF-35 were negative [7, 9]. The diagnosis of synovial sarcoma and malignant peripheral sheath tumor was also excluded because S-100 was negative [10]. Finally, we confirmed the diagnosis of UPS based on the histological findings of HE staining and these excluded diagnoses. (Line84-93)

In the limitation part;

The present study had some limitations. First, the excluded diagnoses were due to the absence of an in-situ hybridization specific test for each tumor [32, 33, 34].

However, we were able to make a definitive diagnosis with HE staining and immunostaining.

(Line172-175)

- PET-CT is not a standard procedure for STS. Moreover, Figure 1 shows a heterogeneous FDG uptake suggesting large necrotic areas rather than low metabolic activity. TEP-FDG is not a gold standard procedure to rule out bacteraemia (as specified in the conclusion)

Author’s response:

Thank you for pointing out this contradiction.

PET was used to look for tumor activity and the presence of metastases. As you pointed out, it was not used to look for bacteremia. Also, you have correctly pointed out that heterogeneous FDG uptake suggested large necrotic areas rather than low metabolic activity.

Author’s action:

We revised the sentence in the conclusion part as shown below:

IUPS can be diagnostically confused with other infections, such as bacteremia, and if there is no indication of bacteremia, blood cultures should be collected immediately to rule out infection. (Lines 172-175)

We also revise the sentences in the discussion part as shown below;

High uptake of FDG indicates higher tumor cellularity, biological behaviors of the tumor cells, and the composition and proportion of inflammatory cells [22]. Moreover, FDG uptake indicates marked leukocytosis in IUPS [5]. In the current case, although the patient showed a high level of inflammation, we observed a relatively low SUV-max; the mean SUV-max value, in general, is reported to be 10.78 ± 6.72, as described previously [23]. The heterogeneous FDG uptake suggested large necrotic areas. (Line151-158)

- The reason why IL6 immunostaining is unclear. Systemic inflammation should be associated to high systemic IL6 serum levels. Was this test performed in this case? Any clues for a link between high systemic IL6 serum levels and tumor expression?

Author’s response:

Thank you for pointing out this mistake. We agree that we should have looked at Il-6 in the blood. Local observations are insignificant.

Therefore, I have added it to the limitation as a prospect for future research.

Author’s action:

We added the sentences in the limitation part as shown below;

In addition, we did not measure the blood levels of systemic inflammatory cytokines such as IL-6.

However, we were able to suggest the possibility of an association between tumor microenvironment and systemic inflammation from the histopathology. This will be an interesting subject for future research. (Line 176-180)

We also revised the sentences in the discussion part as shown below:

In this case, intratumoral infiltration of megakaryocytes and lymphocytes was observed. Thus, it was suggested that inflammatory cytokines might be acting on the tumor microenvironment. (Line137-144)

Minor comments

The manuscript needs English language reediting

Units for CRP and blood cells should homogenized in standard format

Author’s response and action:

We agree with your comment that the manuscript needed the English language re-editing, and the units for CRP and blood cells needed to be aligned; therefore, it was revised as follows;

 CRP unit: mg/dL and blood cells: x 103/µL.

Reviewer 2 Report

This is an interesting manuscript

Lines 61 - 63 are not clear. Final diagnosis is not clearly stated. "wide lesion of necrosis" does not make sense. Definitive inadequate margins are not stated in case report description but only in the discussion (lines 137-138). In case report description the final diagnosis of IUPS should be described and the definitive margins stated.

Either in case report description or the discussion the motivation not to re-operate to obtain final clear margins should be discussed and motivated.

Author Response

Thank you very much for reviewing this paper.

I appreciate your comments, which have helped me in making corrections to the document.

Deletions are shown in red and horizontal lines, and additions are shown in blue.

We believe that we have made a better paper.

I hope you will review it again.

Comments and Suggestions for Authors

This is an interesting manuscript

Lines 61 - 63 are not clear. Final diagnosis is not clearly stated. "wide lesion of necrosis" does not make sense. Definitive inadequate margins are not stated in case report description but only in the discussion (lines 137-138). In case report description the final diagnosis of IUPS should be described and the definitive margins stated.

Author’s response:

Thank you very much for pointing this out.

We believe that the description of extensive necrosis does not add any value and that surgical margins should be described in the case presentation part.

The final definitive diagnosis should also be stated.

Author’s action:

We have revised the sentences in the case presentation part as shown below:

The surgical margin of the resected specimen was microscopically positive [6]. (Line: 66-67)

Either in case report description or the discussion the motivation not to re-operate to obtain final clear margins should be discussed and motivated.

You need to describe why you did not do any additional treatment.

If the surgical margins are poor, as you say, then additional surgery or radiation therapy should be considered.

Author’s response and action:

Thank you very much for pointing this out.

The reason is because the patient and his family did not want to.

I have added this to the discussion part as follows:

Despite inadequate surgical margins, additional radiation therapy and surgical resection were not performed because the patient and his family did not wish to do so. (Line 165-167)

Round 2

Reviewer 1 Report

The authors have properly answered to the previous requests

The description of the tumor microenvironment in this case could still be improved, as well as the description of the several immune classes of sarcoma.

The manuscript also requires some editing of English language

Author Response

Responses to Reviewers’ Comments

Responses to comments of Reviewer#1

Author’s response:

Thank you very much for reviewing our paper. We have revised it according to your insightful comments. We believe that the paper has tremendously benefitted from your insights, and has become more informative. We have also improved the English language of the paper. We hope you will review it again.

Comments and Suggestions for Authors

The authors have properly answered to the previous requests

 The description of the tumor microenvironment in this case could still be improved, as well as the description of the several immune classes of sarcoma.

 The manuscript also requires some editing of English language

Author’s response:

Thank you very much for your helpful review of our paper. We agree that the descriptions of the immune classes of sarcoma in the tumor microenvironment were missing. We apologize for this deficiency. We believe that the tumors in this study could be classified into immuno-high (D and E) and highly vascularized (C) groups. Interestingly, if the tumor is classified as E, it might be an indication for PD-1 inhibitor [reference14], which could be a subject of future research.

Furthermore, we agree with you that the English language of our paper needed improvement. Therefore, we have attempted to improve the language of our paper by availing a professional language editing service.

Author’s action:

We have added the following sentences to the discussion section in the revised paper.

Moreover, Petitprez, et al. established an immune-based classification on the basis of the composition of the tumor microenvironment and identified five distinct phenotypes: immune-low (A and B), immune-high (D and E), and highly vascularized (C) [14]. The report also indicated that the class-E group demonstrated improved survival and a high response rate to PD1 blockade with pembrolizumab in a phase 2 clinical trial [14]. (Lines:129—133).

The current case could be further classified as immune-high (D and E) and/or highly vascularized (C) groups. (Lines:138—139).

We also added the following sentences to the limitations in the revised paper.

Moreover, we did not investigate several immune molecules, such as CD3, CD8, CD20, CD34, PD-1, and PD-L1, to confirm the immune-based classification [14]. If this tumor could have been classified as category E, PD-1 inhibitor therapy might have been indicated. (Lines: 179—182).